# *Proteonematalycus wagneri* Kethley reveals where the opisthosoma begins in acariform mites

**Samuel J. Bolton**<sup></sup>*

Florida State Collection of Arthropods, Division of Plant Industry, Florida Department of Agriculture and Consumer Services, Gainesville, Florida, United States of America

* samuel.bolton77@gmail.com

## Abstract

It is generally thought that the anterior border of the opisthosoma of acariform mites is delineated by the disjugal furrow, but there is no evidence to support this other than the superficial appearance of tagmosis in some oribatids. It is proposed herein that the disjugal furrow is an apomorphic feature that does not correspond with any segmental borders. Although the disjugal furrow is absent from *Proteonematalycus wagneri* Kethley, the visible body segments of this species indicate that this furrow, when present, intersects the metapodosoma. Therefore, the disjugal furrow does not delineate the anterior border of the opisthosoma. Instead, this border is between segments *D* and *E* (segments VI and VII for all arachnids). This hypothesis can be accommodated by a new model in which the proterosoma warps upwards relative to the main body axis. This model, which is applicable to all Acariformes, if not all arachnids, explains the following phenomena: 1) the location of the gnathosomal neuromeres within the idiosoma; 2) the relatively posterior position of the paired eyes; 3) the shape of the synganglion; 4) the uneven distribution of legs in most species of acariform mites with elongate bodies.

## Introduction

Mites comprise two superorders: Parasitiformes and Acariformes. Almost all mites have lost the furrows that delineate their body segments. However, the dorsal setae of acariform mites are arranged in transverse rows, which correspond with the underlying body segments [1]. There are conflicting interpretations concerning the homology of two of these segments, *C* and *D* (bearing setae *c* and *d*). Reuter [2] homologized segments *C* and *D* with the metapodosoma (segments V and VI for all arachnids [1]), which bears legs III and IV. More recently, the same interpretation was put forward by Weigmann [3] (Fig 1B). Van der Hammen [4] instead homologized segments *C* and *D* with the first two segments of the opisthosoma (segments VII and VIII) (Fig 1C and 1D), which was based on his hypothesis that the anterior border of the opisthosoma is represented by the disjugal furrow (Fig 1E). Many, if not most, acarologists accord with van der Hammen's hypothesis by referring to the structures associated with segments *C* and *D* (principally setae and plates) as 'opisthosomal'. The alternative term

**Funding:** The author received no specific funding for this work.

**Competing interests:** The authors have declared that no competing interests exist.

**Fig 1. Terminology and models.** A. Simplified representation of segmentation in mites, showing the segments in order but without any of the hypothesized modifications in shape. B. Interpretation of Weigmann [3]. C. Interpretation of Grandjean [5]. D. Interpretation of Klompen *et al.* [6] E. The three main body furrows of acariform mites. Roman numerals on the segments represent the segmental scheme commonly used for all arachnids [1]. Blue = proterosoma; yellow = metapodosoma; brown = opisthosoma; grey = neutral (no homologue applied); Pc = precheliceral region (acron); Ch = chelicera; Pa = palp; LI–LIV = legs I–IV; white arrowhead = abjugal furrow; black arrowhead = disjugal furrow; grey arrowhead = sejugal furrow.

'hysterosomal' (the region of the body that is behind the sejugal furrow), is neutral with respect to either hypothesis.

Grandjean [5] based a widely cited model on van der Hammen's hypothesis to explain the body segmentation of acariform mites. According to this model, the dorsal region of the podosoma (segments III–VI) is dramatically reduced, causing the parts of the body anterior and posterior to the podosoma to be pulled into areas that were occupied by the podosoma (Fig 1C). Klompen *et al.* [6] also adhered to van der Hammen's hypothesis when they suggested an

amendment to Grandjean's model. They infer that the whole of the prosoma is somewhat evenly warped (Fig 1D), so that the dorsum of the podosoma is again represented as dramatically reduced. These interpretations are illustrated with an animation, which is accessible via the following link: https://zenodo.org/record/5512807#.YUOPiflKhaQ.

Perhaps the main reason that van der Hammen's hypothesis has been so widely adopted is that it adds two additional segments to the acariform body (segments *C* and *D* are treated as separate from the metapodosoma), which makes the body segment count of this lineage closer to that of other arachnids [4]. However, this is not a strong argument because there is noticeable variation in the number of body segments among other arachnids; Scorpiones have nineteen body segments, whereas Opiliones have only fifteen [1].

*Proteonematalycus wagneri* Kethley, a rare species of mite that is only known from sandy habitats from the USA [7], is exceptionally useful for investigating the arrangement of body segments in Acariformes. Unlike other basal acariform mites, in which the furrows that delimit the segments are restricted to the dorsum (best exemplified by Terpnacaridae), some of the furrows of *P. wagneri* completely encircle the hysterosoma, clearly revealing which segments are associated with the metapodosoma. This mite was examined with a scanning electron microscope (SEM) and a light microscope.

## Results

The integument of *P. wagneri* is extremely soft and fragile, causing it to readily distort when it is removed from alcohol. Alcohol stored specimens of this species completely shriveled up when viewed with cryo-SEM, which has been used with a high degree of success on other soft-bodied mites [8]. By comparison, specimens of *P. wagneri* that were desiccated using hexamethyldisilazane (HMDS) provided relatively good images under conventional SEM, although there was still some shriveling (Fig 2A and 2B). SEM reveals that each hysterosomal segment is clearly delimited by intersegmental furrows, which fall on either side of a transverse row of setae. A closeup of the integument shows that the striae break up into very fine protuberances along the base of each furrow (Fig 2B). By revealing furrows that correspond with the borders of all the hysterosomal segments, SEM removes any remaining ambiguity concerning the homology of the furrows with segmental borders. It should be noted that the drawings in the original species description do not include the furrow between segments *F* and *H* [7]. Despite the relatively low image resolution compared to that of SEM, light microscopy demonstrates how *P. wagneri* appears when it has not undergone any shriveling. This species reveals a series of very distinct intersegmental furrows (Fig 2C).

*Proteonematalycus wagneri* shows that segments *C* and *D* bear legs III and IV, respectively, and so these segments represent the metapodosoma (Fig 2A). There is no trace of a disjugal furrow (see Discussion). Note that the borders that are highlighted with small white arrowheads (Fig 2A) represent the dorsolateral edges of coxae III and IV.

## Discussion

### Segmental homology

The presence of visible body segments, involving intersegmental furrows, is very clear in some soft-bodied basal acariform taxa [9, 10]. This is also the case in other basal arachnid lineages, for example, Mesothelae within Araneae [11, 12], and Opilioacarida within Parasitiformes [6, 13]. It is therefore appropriate to base the interpretation of acariform body segmentation on *P. wagneri*, which is a basal acariform mite with a complete set of intersegmental furrows along the hysterosoma.

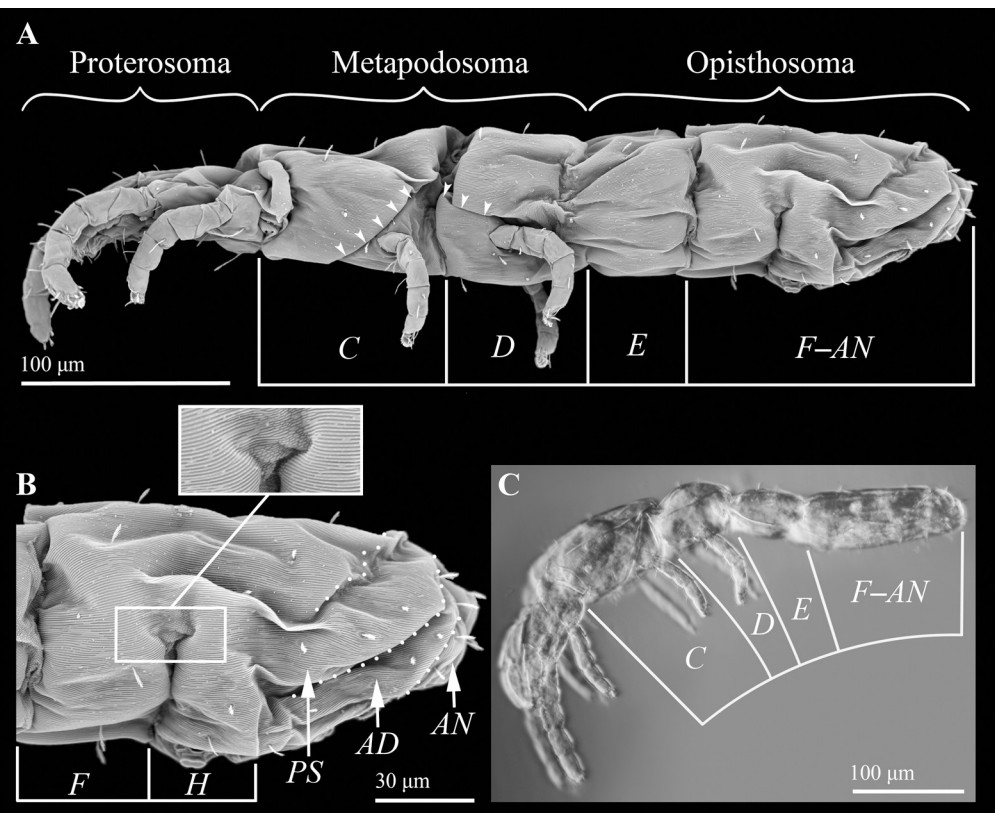

**Fig 2.** *Proteonematalycus wagneri* **Kethley.** A. Lateral view of female (SEM; FSCA 00030222). B. Lateral view of segments *F–AN* (SEM; same female as above). C. Lateral view of female, approximately 2 minutes after immersion in PVA (light microscopy; FSCA 00030224). Small white arrowheads point to the dorsolateral borders of coxae III and IV (note that there is slight sagging of the integument over the border of coxa III). White dotted lines delineate the borders of segments *PS–AN*.

The segmental homology of Acariformes has been problematical because almost all species show no visible trace of their underlying segments other than transverse rows of setae. Segments *C–E* (V–VII) bear the transverse rows of setae *c–e*, which can be readily homologized throughout Acariformes (with a number of exceptions due to hypertrichy). However, these transverse rows of setae are limited to the dorsum, and so it is almost always impossible to confidently determine the ventral extent of the borders of segments *C–E*. In some of the few acariform species where intersegmental furrows are exhibited, e.g., Terpnacaridae, those furrows do not extend to the venter of the hysterosoma.

But for segments *C–E* of *P. wagneri*, the entirety of each of the segmental borders are visible as furrows. Therefore, it is possible to use this very unusual feature of *P. wagneri* to determine the boundaries of segments *C–E* for Acariformes. *P. wagneri* reveals that segments *C* and *D* are metapodosomal (associated with legs III and IV), whereas segment *E* represents the first segment of the opisthosoma (Fig 2A). Mites in the family Tarsocheylidae also show this, although they have an incomplete set of intersegmental furrows along the hysterosoma [14]. These taxa affirm what has already been inferred from other taxonomic groups within Acariformes, namely Siteroptidae [2] and Oribatida [3].

## The disjugal furrow

Van der Hammen [4] used the presence of the disjugal furrow in some oribatids to infer that the dorsum of the metapodosoma is dramatically reduced across all mites. However, there is

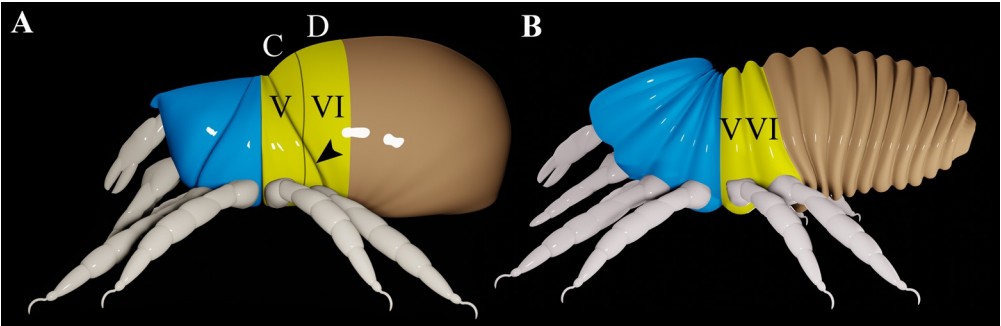

**Fig 3. The metapodosoma of mites.** A. Same model as Fig 1E, but color coded to show that the disjugal furrow intersects the metapodosoma (yellow). B. Opilioacarida, based on Klompen *et al.* [6]. Roman numerals on the metapodosomal segments represent the segmental scheme commonly used for all arachnids [1]. Blue = proterosoma; yellow = metapodosoma; brown = opisthosoma; black arrowhead = disjugal furrow. Thin black line delineates the border between segments *C* and *D*.

no unambiguous evidence for the existence of a disjugal furrow outside of Oribatida. In *Neognathus*, longitudinal ridges of tubercles have been mislabeled as disjugal (each ridge cuts between setae *c1* and *c2* and so cannot delineate the anterior border of the *C* segment) [15]. And based on van der Hammen's hypothesis, the term 'disjugal' is often applied to the dorsal part of the sejugal furrow of non-oribatid taxa [16–19].

The disjugal furrow runs from the front of segment *C* to behind legs IV. This furrow is absent in *P. wagneri*, but if it were present, it would have to cross over a vertical furrow that delineates the border between segments *C* and *D* (Fig 2A and 2C). Therefore, rather than form the boundary between the metapodosoma and prosoma, the disjugal furrow must intersect metapodosomal segments *C* and *D* (segments V and VI) (Fig 3A). And so, this furrow is an apomorphic feature that does not correspond with any segmental border. The disjugal furrow probably evolved as part of a suite of defensive modifications. Oribatids that have this feature are highly modified mites that have undergone a large degree of sclerotization compared to more basal lineages [20, 21]. It is relatively common for border-like structures to form in association with sclerotization. For example, labidostommatids have a distinctive groove that divides the body into a dorsal and ventral shield [22].

Clearly, the disjugal furrow is not evidence that all mites have a metapodosoma in which the dorsum is dramatically reduced relative to the venter. Indeed, no mites appear to have this modification. *Proteonematalycus wagneri* indicates that no such modification has occurred within Acariformes. In most of the main lineages of parasitiform mites the body segmentation is highly ambiguous. The single exception is Opilioacarida. Whereas the dorsum of the metapodosoma of this lineage was thought to be dramatically reduced [23], a more recent interpretation represents this reduction as relatively slight [6] (Fig 3B) (but note that Fig 6 of the same publication [6] appears to show no reduction at all).

## A new model

Mites have been viewed as a distinctive lineage of arachnids based on the presence of a gnathosoma [24, 25], which represents an integrated mouthpart complex that articulates against the idiosoma (rest of the body). However, the gnathosoma is a pseudotagma because the associated neuromeres, the deutocerebrum and tritocerebrum, are located within the idiosoma [26]. Nonetheless, the body segments that are associated with the gnathosoma, segments I and II, are often represented as being anterior to the idiosoma [1, 3]. Accordingly, the neuromeres would have to have migrated, posteriorly, into the idiosoma [26]. But an alternative

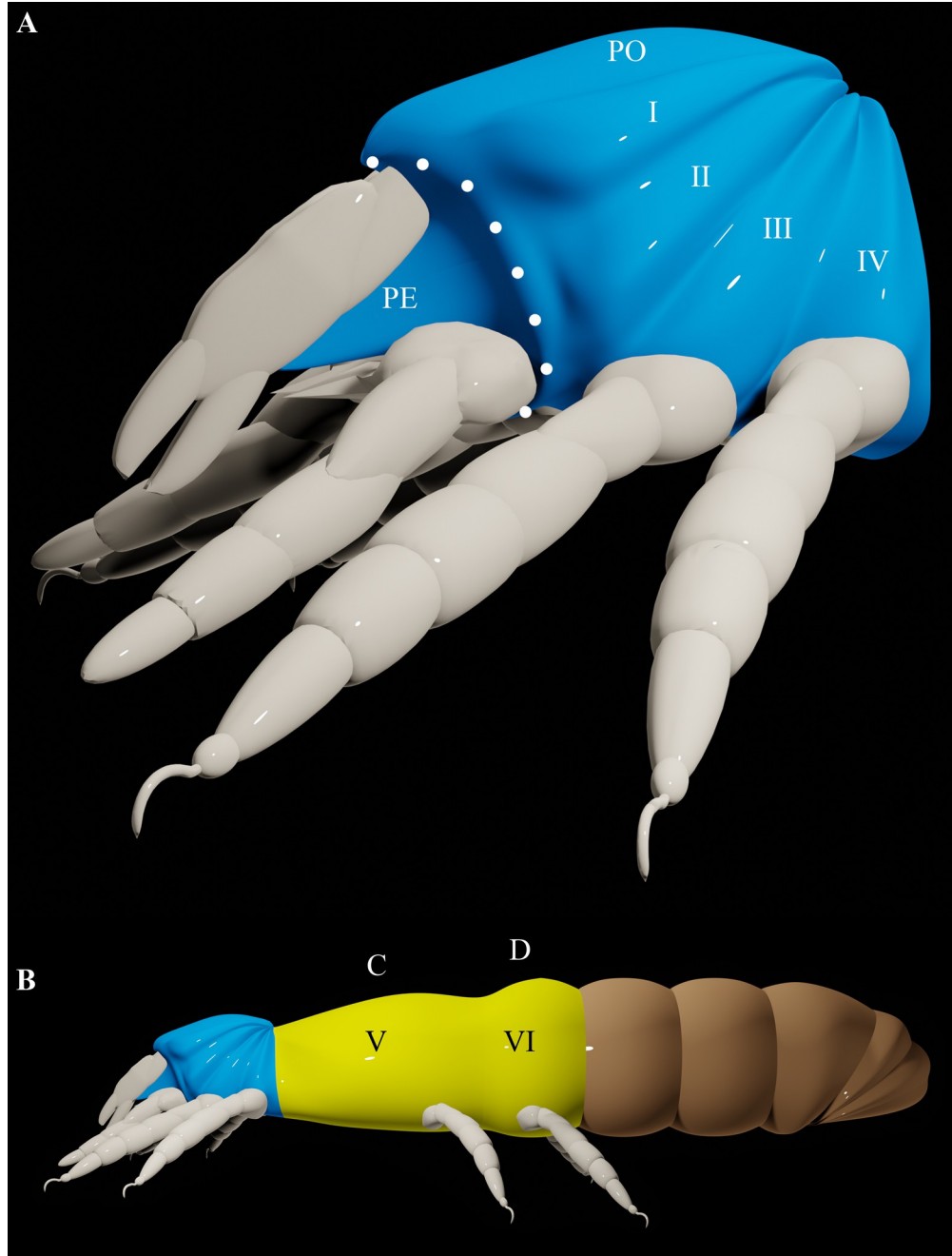

**Fig 4. A new model.** A. The proterosoma (anterolateral view). B. The body of an elongate bodied mite. Roman numerals on the metapodosomal segments represent the segmental scheme commonly used for all arachnids [1]. Blue = proterosoma; yellow = metapodosoma; brown = opisthosoma; PE = precheliceral region—epistomo-labral projection; PO = precheliceral region—ocular (also prodorsum or propeltidium); white dotted line delineates the circumcapitular furrow (boundary between gnathosoma and idiosoma).

explanation for the relatively posterior position of the neuromeres is that the proterosoma warps upwards, relative to the main body axis, so that the gnathosoma is anterior to segments I and II (Fig 4A). These body segments, which contain the deutocerebrum and tritocerebrum,

are part of the idiosoma rather than the gnathosoma. This would mean that the circumcapitular furrow, which delineates the boundary between the gnathosoma and idiosoma, would not correspond with any segmental border (except possibly the ventral part of the border between segments II and III). This is also a necessary implication of two of the other three main models [5, 6] (Fig 1C and 1D). Indeed, according to van der Hammen [27], the gnathosoma is a pseudotagma that is secondarily articulating, which implies that the circumcapitular furrow is not segmental in origin.

Weigmann [3], Grandjean [5] and Klompen *et al*. [6] also proposed that the anterior region of the body is warped (Fig 1B–1D). According to Weigmann, only the gnathosoma is warped, but this fails to explain why the associated neuromeres are located within the idiosoma. This also does not explain the presence of eyes on the prodorsum or dorsal shield in many mites (see below). According to Grandjean and Klompen *et al*., the entire prosoma is warped. The metapodosoma is unwarped in the new model that is proposed herein (Fig 4B). This distinction is important because the region and extent of warping should have an effect on how body elongation proceeds in mites. If only the proterosoma is warped (in accordance with this new model), elongation of any of the proterosomal segments would extend the proterosoma along a vertical or oblique axis relative to the main axis of the body, causing the body to form a kink rather than become more elongated. But no such kink would arise to the body when the metapodosoma elongates because this part of the prosoma is aligned with the main axis of the body. Any dramatic elongation of the prosoma must therefore proceed via the metapodosoma rather than the proterosoma. This can explain the uneven distribution of legs in most species of acariform mites with elongate bodies (Fig 4B). Legs I and II are always tightly packed with the palps and chelicerae because the proterosoma does not elongate. But legs III and IV can shift to a much more posterior position because the metapodosoma is free to elongate [14, 28, 29] (Fig 2).

Warping of the entire prosoma or only of the gnathosoma cannot explain the uneven distribution of legs nearly as well. If the whole of the prosoma is warped, dramatic elongation should only occur to the part of the body that is posterior to legs IV, but this is clearly not the case. And if only the gnathosoma is warped, legs I and II should not always be tightly packed with the palps and chelicerae.

Upward warping of the proterosoma appears to be a feature of all arachnids. It would cause the part of the precheliceral region that bears the eyes to lie directly above the rest of the proterosoma, including the body segments that bear legs I and II. This can explain why various lineages, including Acariformes, Opilioacarida, Opiliones and Scorpiones, have eyes that are in a relatively posterior position on the prosoma, directly above the leg coxae. Moreover, the region of the synganglion that is associated with the proterosoma is also warped [30–33]. This part of the synganglion comprises the supraesophageal section and the first two neuromeres (associated with legs I and II) of the subesophageal section. The posterior part of the subesophageal section, which is associated with the metapodosoma, is more closely aligned with the main body axis. Therefore, the proterosomal warp may be a feature that is fundamental to the organization of the arachnid body. It is noteworthy that, primitively, the euarthropod is considered to have undergone tagmosis into a head (proterosoma) and trunk (hysterosoma) [34, 35]. Warping of the proterosoma may in some way be linked to this tagmosis.

A commonly held view of the prosoma of acariform mites is that the body segments that are associated with the gnathosoma, segments I and II, are anterior to the prodorsum, which would accordingly form the dorsum of segments III and IV [3, 36]. Consequently, the prodorsum would have no known homologue in other arachnids [1]. But if the proterosoma is warped, segments I and II cannot be anterior to the prodorsum. Instead, the prodorsum lies directly above these body segments (Fig 4A), which means that, sequentially, the prodorsum comes before them. Therefore, the prodorsum would be homologous with both the

precheliceral region (excluding the labrum) and the propeltidium (the dorsal shield of the proterosoma), a structure that is found in Palpigradi, Solifugae and Schizomida.

In accordance with this new model, the proterosoma is warped and the metapodosoma is parallel, relative to the main body axis. There appears to be no arachnid that shows clear evidence that the metapodosoma is warped, whereas a warped proterosoma appears to be a feature of all arachnids. Therefore, this new model may be applicable to all arachnids.

## Materials and methods

### Collection

*Proteonematalycus wagneri* was collected from foredune sand using heptane flotation [37]. Collection event: U.S.A., Indiana, Porter Co., Indiana Dunes State Park, 41.6780 N 87.0081 W, sand dune (10 cm deep); collector: Samuel Bolton, 16 May, 2013 (FSCA 00030222: Female x1, Deutonymph x1, Protonymph x1, larva x1) (FSCA 00030224: Female x1; FSCA 00030225: Deutonymph x1).

### SEM

Specimens of *P. wagneri* were transferred from a storage medium of 95% ethanol into the following series of solvents: 1) absolute ethanol; 2) 50:50 volume HMDS; 3) 100% HMDS. Immersion in each fluid medium lasted approximately five minutes. A minuten pin, which had been glued onto a FisherbrandTM plain wooden applicator, was used to maneuver the specimens between solutions. For the final step, the HMDS was left to evaporate. The specimens were then mounted on SEM stubs and sputter coated with approximately 70 nm of gold/palladium using a Denton IV sputtercoater. Micrographs were captured with a Phenom XL G2 Desktop SEM.

### Light microscopy

Examination was with a compound microscope (Leica DM2500) equipped with differential interference contrast (DIC) and a digital SLR (Canon EOS 80D). Imaging was with a dry, 10x objective (brightfield). Polyvinyl alcohol (PVA) mounting medium was used to hold the specimens in a relatively still position. Specimens were imaged without a coverslip within minutes of being placed into the PVA. Contrast was heightened by setting the turret of the DIC between the 10 and 40 intervals.

## Supporting information

**S1 Fig. Larva FSCA 00030222.**
(TIF)

**S2 Fig. Protonymph FSCA 00030222.**
(TIF)

**S3 Fig. Protonymph FSCA 00030222.**
(TIF)

**S4 Fig. Deutonymph FSCA 00030222.**
(TIF)

**S5 Fig. Adult FSCA 00030222.**
(TIF)

**S6 Fig. Deutonymph FSCA 00030225 (eight minutes after immersion in PVA).**
(TIF)

**S7 Fig. Adult FSCA 00030224 (two minutes after immersion in PVA).**
(TIF)

## Acknowledgments

Jonathan Bremer, at the Florida State Collection of Arthropods, undertook the SEM imaging of *P. wagneri* (Fig 2A and 2B). The U.S. National Park Service granted the author permission to collect from the Indiana Dunes National Lakeshore. The following individuals are thanked for their contribution through internal review and/or useful discussion: Erin Powell, Elijah Talamas, Paul Skelley (Florida Department of Agriculture and Consumer Services, Gainesville, Florida, USA) and Hans Klompen (Ohio State University, Columbus, Ohio, USA). The Florida Department of Agriculture and Consumer Services–Division of Plant Industry are thanked for their support on this contribution.

## Author Contributions

**Writing – review & editing:** Samuel J. Bolton.

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
