## [Decision Letter · Decision Letter 0]

5 Nov 2021

PONE-D-21-31633Proteonematalycus wagneri Kethley reveals where the opisthosoma begins in acariform mitesPLOS ONE

Dear Dr. Bolton,

Thank you for submitting your manuscript to PLOS ONE. After careful consideration, we feel that it has merit but does not fully meet PLOS ONE’s publication criteria as it currently stands. Therefore, we invite you to submit a revised version of the manuscript that addresses the points raised during the review process.

Please pay particular attention to the comments, concerns, and questions raised by the reviewers in their comments. In general, they will strengthen the manuscript by clearing up confusion and covering alternative points of view. Both reviewers were complimentary of the work and appreciative of the contribution. I generally agree with their positions.

We look forward to receiving your revised manuscript.

Kind regards,

Michael Scott Brewer, Ph.D.

Academic Editor

PLOS ONE

Journal Requirements:

Reviewers' comments:

Reviewer's Responses to Questions

**Comments to the Author**

1. Is the manuscript technically sound, and do the data support the conclusions?

Reviewer #1: Yes

Reviewer #2: Yes

2. Has the statistical analysis been performed appropriately and rigorously? 

Reviewer #1: N/A

Reviewer #2: N/A

3. Have the authors made all data underlying the findings in their manuscript fully available?

Reviewer #1: Yes

Reviewer #2: Yes

4. Is the manuscript presented in an intelligible fashion and written in standard English?

Reviewer #1: Yes

Reviewer #2: Yes

5. Review Comments to the Author

Reviewer #1: Arachnids conventionally have a body divided into an anterior prosoma (or cephalothorax) and posterior opisthosoma (or abdomen), but in some groups the two body halves can fuse together to some extent which makes this fundamental division less obvious. The author has submitted an interesting manuscript which goes some way towards clarifying a long-standing problem of how to homologise the (reduced) body of acariform mites with those of other arachnids: in particular where does the opisthosoma begin? The author should be congratulated for producing high-quality images of a soft-cuticled species which I'm sure was challenging to handle, and makes a good case that the leg-bearing segments conventionally referred to as C & D belong to the metapodosoma part of the prosoma. The opisthsoma, in turn, begins behind the segment bearing the last pair of legs as in other arachnids. In this sense the study certainly merits publication and could be suitable for a journal such as PLOS One.

However, because its being submitted to a high-profile interdisciplianary journal (and not a specialist acarology publication) I think the manuscript could benefit from placing these results in a slightly broader context, making more comparisons with Arachnida in general. Essentially your data, as I understand it, suggests that the ground pattern of acariforms is a kind of straightforward 4+2 pattern of prosomal segmentation, immediately followed by the opisthosoma. This would make acariforms anatomically similar to things like schizomids, palpigrades and camel spiders (Solifugae). It may be worth noting that at least palpigrades and camel spiders have been proposed, in some phylogenies, as the putative sister-group of acariform mites.

As I'm sure the author knows there has also been a long history of mites being described using unique morphological terms, often with limited efforts to homologise them with structures in other arachnids. This hinders comparative morphology, and even phylogeny when homologous characters states are obscured by nomenclature. In the present case, would it be worth expressing the different hypotheses in terms of conventional numbering used in other arachnids? In other words, in the scenario of Weigmann and yourself C and D are prosomal segments 5 and 6, but in the other models they are opisthsomal segments 1 and 2 (or body segments 7 and 8). This could perhaps be done by adding segment numbers to Fig. 1A for example.

Related to this, I was thinking to what extent must we seperate dorsal elements from ventral (limb-bearing) elements. In lines 32 you say "There are conflicting interpretations concerning the relationship of these two segments, C and D...to the metapodosoma, which bears legs III and IV." I think what I'm getting at is which segments bear legs III and IV in the Van Der Hammen / Grandjean / Klompen scenarios? It can't be C and D, which they interpret as opisthosomal segments 1 and 2, so what, if anything, did they call the ventral metapodosomal elements bearing the last pair of limbs?

Another problem is that I couldn't wholly follow how you go from Figure 2A to Figure 3B, with the disjugal suture of oribatids cutting across one (or more) segments of the metapodosoma? You state that P. wagneri lacks a disjugal suture, which I'm sure from your photos is correct. I think the problem lies with lines 88-92 which should include a bit more detail about why the disjugal suture cannot be the typical arachnid prosoma/opisthosoma boundary, and what exactly you mean when you define a suture as something which only intersects a plane. How do you recognise this in actual specimens of mites? You propose that an intersegmental furrow is associated with a narrowing, but (to play devil's advocate) the disjugal suture in your idealised Fig. 3C could be seen as creating a slight fold or narrowing between the regions in front of and behind it. So why is it a suture (other than because of its traditional name?). In essence, your folded suture in Fig. 3B does not look like your smooth suture in Fig. 3A which, if anything, looks more like the prosoma-opisthosoma boundary in 3A.

Also, I did wonder if there is any embryological or hox gene work to support your model? Do the papers by Richard Thomas and/or Austen Barnett help? I recall that they found evidence for only 2 unequivocal segments in the oribatid opisthosoma, but did they indicate where (anatomically) the opisthosoma begins?

https://onlinelibrary.wiley.com/doi/abs/10.1111/j.1525-142X.2012.00556.x

https://evodevojournal.biomedcentral.com/articles/10.1186/2041-9139-4-23

https://www.proquest.com/openview/f22b93c07a71a645657915c444208a67/1?pq-origsite=gscholar&cbl=18750

Again, the absence of segment numbers (or even the C, D, E scheme) makes it a little hard to follow the argumentation through into the higher oribatids and my question is, in the 3B model where (if anywhere) is the boundary between the segments bearing legs III and IV? The model implies it is not the disjugal suture, as this appears to originate behind leg IV, so have C and D (= 5 and 6) fused together here into an undifferentiated metapodosoma with (in your scenario) the disjugal suture arising as a novel structure and cutting across it?

The video of the body plan morphing into different hypotheses is nice. I'm not sure how much extra work is involved, but if they colour scheme used in Figure 1 were to be added it would be even clearer where different authors interpreted the start of the opisthosoma.

Other minor corrections

REFERENCES

May be a formatting problem, but genus and species names need to be italicised here throughout.

FIGURES

Figure 1: There is no part 'D' in the figure legend, but two part Es. I think it should read "D. Interpretation of Klompen et al. [6]"

Figure 2B: What is being highlighted by the white box? A furrow or protubrence? This is not really explained in the figure legend.

Reviewer #2: I am no expert in arthropod or mite segmentation, but this short paper appears to make a clear point about the need to adjust certain boundaries in body parts, namely the true border between opisthosoma and metapodosoma. The paper is straight to the point, clearly written and fairly strongly argued. The 2 main figures are extremely simple and clear (despite the complexity behind it), and the supplementary figs are also quite useful to support his hypothesis.

The video is also neatly done, appealing and can reinforce understanding. I suggest to keep this link in the paper.

See the main PDF for a few parts that need minor adjustments, mostly for improving clarity.

Otherwise, I see a few potential ambiguities that need clarifying:

-Abstract, line 17: you say that the sejugal suture intersects/crosses the metapodosoma in P. wagneri. But later you say that the disjugal suture is probably an apomorphy only present in oribatids. This appears as a contradiction. Should we or should we not consider that a sejugal suture is present in mites other than oribatids? It seems important to clarify this here.

-Your fig. 3B and text indicate that the sejugal suture is not concordant/homologous to the border between prosoma and opisthosoma, and that the metapodosomal-opisthosoma border is somewhere posterior to the sejugal suture in the mites having a distinct sejugal suture. Should you give further considerations for the mites having a clear sejugal suture (e.g. Oribatida), such as explaining that since the homology of the series of c and d setae is established across Acariformes (e.g. oribatids and trombidiforms), therefore it is clear that mites with sejugal suture have their metapodosoma border somewhere between the d and e setae?

If appropriate, there could be a small concluding paragraph, that would include points such as: the Weigman model can be applied to many acariform mites? And what are the main consequences of your finding for taxonomic and/or developmental biology sciences? Is there any consequence other than that segments C-D and their setae, born dorsally (c,d), should be considered as part of the metapodosoma (not the opisthosoma)?

More specific comments:

Line 88: "no real resemblance" is a bit ambiguous, because the sejugal suture is somehow between 2 dorsal humps, therefore it can be percieved/concieved as a furrow, although not intersegmental furrow, based on your findings. So, I suggest to add nuances to your sentence.

Fig 2A: some readers may at first glance wonder if the dorsolateral border of coxal field IV could represent the lower part of the disjugal suture. I suggest to mention these borders (ofboth coxal field III and IV) in your Results to avoid any possible confusion?

Fig 1 caption:

There are 2 E listed. Change the first E for D?

Fig 3 caption. I suggest to add “...for explaining segmentation in elongate...”

6. PLOS authors have the option to publish the peer review history of their article (what does this mean?). If published, this will include your full peer review and any attached files.

Reviewer #1: **Yes: **Jason A. Dunlop

Reviewer #2: No

---

## [Author Response · Author response to Decision Letter 0]

6 Dec 2021

Dear Editor,

Although this is supposedly a letter of rebuttal, I see almost nothing to rebut. I think I have never had such constructive and useful feedback from a pair of reviewers. And although this paper is supposed to require only minor amendments, I found myself having to make some fairly major changes based on the insightful responses of the reviewers (see below).

Best regards,

Sam

-----

Reviewer #1: Arachnids conventionally have a body divided into an anterior prosoma (or cephalothorax) and posterior opisthosoma (or abdomen), but in some groups the two body halves can fuse together to some extent which makes this fundamental division less obvious. The author has submitted an interesting manuscript which goes some way towards clarifying a long-standing problem of how to homologise the (reduced) body of acariform mites with those of other arachnids: in particular where does the opisthosoma begin? The author should be congratulated for producing high-quality images of a soft-cuticled species which I'm sure was challenging to handle, and makes a good case that the leg-bearing segments conventionally referred to as C & D belong to the metapodosoma part of the prosoma. The opisthsoma, in turn, begins behind the segment bearing the last pair of legs as in other arachnids. In this sense the study certainly merits publication and could be suitable for a journal such as PLOS One. However, because its being submitted to a high-profile interdisciplianary journal (and not a specialist acarology publication) I think the manuscript could benefit from placing these results in a slightly broader context, making more comparisons with Arachnida in general. Essentially your data, as I understand it, suggests that the ground pattern of acariforms is a kind of straightforward 4+2 pattern of prosomal segmentation, immediately followed by the opisthosoma. This would make acariforms anatomically similar to things like schizomids, palpigrades and camel spiders (Solifugae). It may be worth noting that at least palpigrades and camel spiders have been proposed, in some phylogenies, as the putative sister-group of acariform mites.

Author: Thank you greatly for this suggestion. This has caused me to make some fairly big changes to the manuscript, and so the paper has been somewhat revamped. I have gone to some effort to expand the model section of the paper in order to broaden the context of the chief finding. I originally had major reservations about extending this model to all arachnids. I am simply not as confident on the rest of Arachnida. But I don’t see that any arachnid deviates from the model that I am proposing. However, if I am going to put this interpretation of the prosoma of mites into the context of all arachnids, I think I have to tackle a misunderstanding that has arisen with respect to the gnathosoma. The circumcapitular furrow is often treated as a segmental border, but this would mean that mites are completely different from other arachnids. This is where I find myself agreeing with van der Hammen. He recognized that the circumcapitular furrow is not a segmental border, which means that the body segments associated with the gnathosoma are not within the gnathosoma itself. And so, I am to some extent according with van der Hammen on the front end of the prosoma, and I am going against him on the back end of the prosoma. Hopefully all of this is a lot clearer in the text while also putting mites into a broader context. I must admit that I have shied away from talking about the phylogenetic significance of a propeltidium, although I accept that there is a fairly high likelihood that Solifugae are sister to either Acari or Acariformes. My general feeling is that this character is too evolutionarily plastic to be very phylogenetically informative for reconstructing phylogenetic relationships among arachnids, especially because it is also found in Schizomida. Although I could discuss these doubts, I would prefer to keep this section as brief as possible because it is side tracking from the main thrust of the paper. 

-----

Reviewer #1: As I'm sure the author knows there has also been a long history of mites being described using unique morphological terms, often with limited efforts to homologise them with structures in other arachnids. This hinders comparative morphology, and even phylogeny when homologous characters states are obscured by nomenclature. In the present case, would it be worth expressing the different hypotheses in terms of conventional numbering used in other arachnids? In other words, in the scenario of Weigmann and yourself C and D are prosomal segments 5 and 6, but in the other models they are opisthosomal segments 1 and 2 (or body segments 7 and 8). This could perhaps be done by adding segment numbers to Fig. 1A for example.

Author: This is now addressed in Fig. 1 A–D and in the associated text of the introduction. 

-----

Reviewer #1: Related to this, I was thinking to what extent must we seperate dorsal elements from ventral (limb-bearing) elements. In lines 32 you say "There are conflicting interpretations concerning the relationship of these two segments, C and D...to the metapodosoma, which bears legs III and IV." I think what I'm getting at is which segments bear legs III and IV in the Van Der Hammen / Grandjean / Klompen scenarios? It can't be C and D, which they interpret as opisthosomal segments 1 and 2, so what, if anything, did they call the ventral metapodosomal elements bearing the last pair of limbs?

Author: This is now addressed through the same changes mentioned above. The segments bearing legs III and IV are segments V and VI.

-----

Reviewer #1: Another problem is that I couldn't wholly follow how you go from Figure 2A to Figure 3B, with the disjugal suture of oribatids cutting across one (or more) segments of the metapodosoma? 

Author: This is now addressed in the second paragraph of the section on the disjugal furrow. The disjugal suture was addressed in lines 88-92 in order to define the difference between a suture and a furrow, but that part is now deleted (see below). 

-----

Reviewer #1: You state that P. wagneri lacks a disjugal suture, which I'm sure from your photos is correct. I think the problem lies with lines 88-92 which should include a bit more detail about why the disjugal suture cannot be the typical arachnid prosoma/opisthosoma boundary, and what exactly you mean when you define a suture as something which only intersects a plane. How do you recognise this in actual specimens of mites? You propose that an intersegmental furrow is associated with a narrowing, but (to play devil's advocate) the disjugal suture in your idealised Fig. 3C could be seen as creating a slight fold or narrowing between the regions in front of and behind it. So why is it a suture (other than because of its traditional name?). In essence, your folded suture in Fig. 3B does not look like your smooth suture in Fig. 3A which, if anything, looks more like the prosoma-opisthosoma boundary in 3A.

Author: The other reviewer also picked up on this problem with reference to line 88. I am fond of keeping papers as succinct as possible and dropping concepts and definitions that can come across ambiguously. I am no longer convinced that there is a very robust way of distinguishing a suture from a furrow. There are sometimes humps on either side of a suture, which can make it almost indistinguishable from a furrow. Therefore, I have dropped any mention of suture from the manuscript and simply switched to using the term furrow, which accords with much of the literature. I have made adjustments where necessary. I think the paper is relatively unscathed by this.

-----

Reviewer #1: Also, I did wonder if there is any embryological or hox gene work to support your model? Do the papers by Richard Thomas and/or Austen Barnett help? I recall that they found evidence for only 2 unequivocal segments in the oribatid opisthosoma, but did they indicate where (anatomically) the opisthosoma begins?

https://onlinelibrary.wiley.com/doi/abs/10.1111/j.1525-142X.2012.00556.x

https://evodevojournal.biomedcentral.com/articles/10.1186/2041-9139-4-23

https://www.proquest.com/openview/f22b93c07a71a645657915c444208a67/1?pq-origsite=gscholar&cbl=18750

Author: I have not cited these papers because there is no mention of the chaetotaxy (c, d and e setae). So, they can’t help with the determination of the beginning of the opisthosoma in relation to the rows of transverse setae.

-----

Reviewer #1: Again, the absence of segment numbers (or even the C, D, E scheme) makes it a little hard to follow the argumentation through into the higher oribatids and my question is, in the 3B model where (if anywhere) is the boundary between the segments bearing legs III and IV? The model implies it is not the disjugal suture, as this appears to originate behind leg IV, so have C and D (= 5 and 6) fused together here into an undifferentiated metapodosoma with (in your scenario) the disjugal suture arising as a novel structure and cutting across it?

Author: I have amended Figure 3B and 3C (now 3A and 4B), so that you can see segments C and D, also labelled segment V and VI.

-----

Reviewer #1: The video of the body plan morphing into different hypotheses is nice. I'm not sure how much extra work is involved, but if they colour scheme used in Figure 1 were to be added it would be even clearer where different authors interpreted the start of the opisthosoma.

Author: This is the one place where I feel the need to resist. The video is meant as a standalone piece, largely to encourage people to read this paper. I prefer not to add any more colors, which would complicate it because then I would feel obliged to add a key. It is aimed at the broadest possible audience and is largely designed to give people a first impression about the competing ideas on the segmental homology of mites.

-----

Reviewer #1: May be a formatting problem, but genus and species names need to be italicized here throughout.

Author: Thanks. This should be corrected now.

-----

Reviewer #1: Figure 1: There is no part 'D' in the figure legend, but two part Es. I think it should read "D. Interpretation of Klompen et al. [6]"

Author: Thank you! This has been corrected.

-----

Reviewer #1: Figure 2B: What is being highlighted by the white box? A furrow or protubrence? This is not really explained in the figure legend.

Author: This is now addressed in the caption of Figure 2. My chief concern is that the PDF provides only very low resolution, which may be why this does not seem at all clear. But this should be much clearer in the final publication.

Reviewer #2: I am no expert in arthropod or mite segmentation, but this short paper appears to make a clear point about the need to adjust certain boundaries in body parts, namely the true border between opisthosoma and metapodosoma. The paper is straight to the point, clearly written and fairly strongly argued. The 2 main figures are extremely simple and clear (despite the complexity behind it), and the supplementary figs are also quite useful to support his hypothesis. The video is also neatly done, appealing and can reinforce understanding. I suggest to keep this link in the paper. See the main PDF for a few parts that need minor adjustments, mostly for improving clarity. Otherwise, I see a few potential ambiguities that need clarifying. Abstract, line 17: you say that the sejugal suture intersects/crosses the metapodosoma in P. wagneri. But later you say that the disjugal suture is probably an apomorphy only present in oribatids. This appears as a contradiction. Should we or should we not consider that a sejugal suture is present in mites other than oribatids? It seems important to clarify this here.

Author: Yes, this was awkwardly phrased in the abstract. It is hopefully now amended so that it is clear. Despite extensive searching, I can find no evidence of a disjugal suture in mites outside of Oribatida. I assume that is what is being referred to. A sejugal furrow, on the other hand, is present in many mites outside of Oribatida.

-----

Reviewer #2: Your fig. 3B and text indicate that the sejugal suture is not concordant/homologous to the border between prosoma and opisthosoma, and that the metapodosomal-opisthosoma border is somewhere posterior to the sejugal suture in the mites having a distinct sejugal suture. Should you give further considerations for the mites having a clear sejugal suture (e.g. Oribatida), such as explaining that since the homology of the series of c and d setae is established across Acariformes (e.g. oribatids and trombidiforms), therefore it is clear that mites with sejugal suture have their metapodosoma border somewhere between the d and e setae?

Author: Thanks very much for this suggestion! A new section has been put together to address this, titled “segmental homology”. This is mostly to explain why a single species of mite can be used to determine the true positions of the C and D setae across all Acariformes. The second paragraph addresses this point about the homology of the c and d setae, along with other related points. I think this new version is now much better thanks to this new section. I specifically address how this affects the disjugal suture in the second paragraph of the section on the disjugal suture (please also see the new Fig. 3A).

-----

Reviewer #2: If appropriate, there could be a small concluding paragraph, that would include points such as: the Weigman model can be applied to many acariform mites? And what are the main consequences of your finding for taxonomic and/or developmental biology sciences? Is there any consequence other than that segments C-D and their setae, born dorsally (c,d), should be considered as part of the metapodosoma (not the opisthosoma)?

Author: I disagree with the Weigmann model because there is no warping of the proterosoma. I don’t think that model can be applied to any mites. But the main thrust of this paper was to address where Weigmann was correct. I have now made my position clearer in the model section of the discussion while also emphasizing that the basic arrangement of the body segments of mites is probably the same as other arachnids. This is the main implication of the new model that is proposed in the paper. This model essentially hybridizes Klompen et al.’s interpretation of Acariformes with that of Weigmann. The new model has greater explanatory power because it can be applied to elongate bodied mites. The other two models do a poor job of explaining the large interval between legs II and III. This is now all detailed in the paper. But there are no really important implications with respect to taxonomy and developmental biology other than what is already included. 

-----

Reviewer #2: Line 88: "no real resemblance" is a bit ambiguous, because the sejugal suture is somehow between 2 dorsal humps, therefore it can be percieved/concieved as a furrow, although not intersegmental furrow, based on your findings. So, I suggest to add nuances to your sentence.

Author: The other reviewer also picked up on this problem. I am fond of keeping papers as succinct as possible and dropping concepts and definitions that can come across ambiguously. I am no longer convinced that there is a very robust way of distinguishing a suture from a furrow. There are sometimes humps on either side of a suture, which can make it almost indistinguishable from a furrow. Therefore, I have dropped any mention of suture from the manuscript and simply switched to using the term furrow, which accords with much of the literature. I have made adjustments where necessary. I think the paper is relatively unscathed by this.

-----

Reviewer #2: Fig 2A: some readers may at first glance wonder if the dorsolateral border of coxal field IV could represent the lower part of the disjugal suture. I suggest to mention these borders (ofboth coxal field III and IV) in your Results to avoid any possible confusion?

Author: Amended. Thanks!

-----

Reviewer #2: Fig 1 caption: There are 2 E listed. Change the first E for D?

Author: Thank you! This has been corrected.

---

## [Decision Letter · Decision Letter 1]

9 Feb 2022

Proteonematalycus wagneri Kethley reveals where the opisthosoma begins in acariform mites

PONE-D-21-31633R1

Dear Dr. Bolton,

We’re pleased to inform you that your manuscript has been judged scientifically suitable for publication and will be formally accepted for publication once it meets all outstanding technical requirements.

Kind regards,

Michael Scott Brewer, Ph.D.

Academic Editor

PLOS ONE

Additional Editor Comments (optional):

I apologize for the delay. One of the reviewers requested an extension but still has not finished the review. I looked through the comments and the way you addressed them. I find them satisfying and am willing to accept the manuscript. The reviewer's original concerns were minor anyway. Thank you for your patience and your great work.

Reviewers' comments:

Reviewer's Responses to Questions

**Comments to the Author**

1. If the authors have adequately addressed your comments raised in a previous round of review and you feel that this manuscript is now acceptable for publication, you may indicate that here to bypass the “Comments to the Author” section, enter your conflict of interest statement in the “Confidential to Editor” section, and submit your "Accept" recommendation.

Reviewer #1: All comments have been addressed

2. Is the manuscript technically sound, and do the data support the conclusions?

Reviewer #1: Yes

3. Has the statistical analysis been performed appropriately and rigorously? 

Reviewer #1: N/A

4. Have the authors made all data underlying the findings in their manuscript fully available?

Reviewer #1: Yes

5. Is the manuscript presented in an intelligible fashion and written in standard English?

Reviewer #1: Yes

6. Review Comments to the Author

Reviewer #1: The author has addressed the previous comments and the revised text and figures are now acceptable for publication. I look forwards to seeing this interesting manuscript published.

7. PLOS authors have the option to publish the peer review history of their article (what does this mean?). If published, this will include your full peer review and any attached files.

Reviewer #1: **Yes: **Jason Dunlop

---

## [Editor Report · Acceptance letter]

16 Feb 2022

PONE-D-21-31633R1 

*Proteonematalycus wagneri* Kethley reveals where the opisthosoma begins in acariform mites 

Dear Dr. Bolton:

I'm pleased to inform you that your manuscript has been deemed suitable for publication in PLOS ONE. Congratulations! Your manuscript is now with our production department. 

Kind regards, 

on behalf of

Dr. Michael Scott Brewer 

Academic Editor

PLOS ONE